# Anticipatory Postural Adjustments and Compensatory Postural Responses to Multidirectional Perturbations—Effects of Medication and Subthalamic Nucleus Deep Brain Stimulation in Parkinson’s Disease

**DOI:** 10.3390/brainsci13030454

**Published:** 2023-03-07

**Authors:** Tobias Heß, Christian Oehlwein, Thomas L. Milani

**Affiliations:** 1Department of Human Locomotion, Chemnitz University of Technology, 09126 Chemnitz, Germany; 2Neurological Outpatient Clinic for Parkinson Disease and Deep Brain Stimulation, 07551 Gera, Germany

**Keywords:** Parkinson’s disease, deep brain stimulation, subthalamic nucleus, postural instability, motor control, anticipatory postural adjustments, compensatory postural responses, multidirectional perturbations

## Abstract

Background: Postural instability is one of the most restricting motor symptoms for patients with Parkinson’s disease (PD). While medication therapy only shows minor effects, it is still unclear whether medication in conjunction with deep brain stimulation (DBS) of the subthalamic nucleus (STN) improves postural stability. Hence, the aim of this study was to investigate whether PD patients treated with medication in conjunction with STN-DBS have superior postural control compared to patients treated with medication alone. Methods: Three study groups were tested: PD patients on medication (PD-MED), PD patients on medication and on STN-DBS (PD-MED–DBS), and healthy elderly subjects (HS) as a reference. Postural performance, including anticipatory postural adjustments (APA) prior to perturbation onset and compensatory postural responses (CPR) following multidirectional horizontal perturbations, was analyzed using force plate and electromyography data. Results: Regardless of the treatment condition, both patient groups showed inadequate APA and CPR with early and pronounced antagonistic muscle co-contractions compared to healthy elderly subjects. Comparing the treatment conditions, study group PD-MED–DBS only showed minor advantages over group PD-MED. In particular, group PD-MED–DBS showed faster postural reflexes and tended to have more physiological co-contraction ratios. Conclusion: medication in conjunction with STN-DBS may have positive effects on the timing and amplitude of postural control.

## 1. Introduction

Parkinson’s disease (PD) is known as a widespread neurodegenerative movement disorder which commonly affects elderly people over the age of 60. Its incidence is rising progressively due to demographic changes, while its cause remains partly unknown [1,2,3]. 

The major underlying pathological mechanism is related to the depletion of dopaminergic neurons within the substantia nigra pars compacta, which causes a hypo-dopaminergic state within the basal ganglia. The lack of sufficient dopamine leads to imbalanced excitatory and inhibitory communication within the nigro-striatal and thalami-cortical circuits and disrupts the interaction of brain regions responsible for selecting, planning, adapting, and executing motor programs [4,5,6,7,8]. It has been shown that after losing about 80% of dopamine-producing cells, patients initially experience predominantly motor-based symptoms, such as tremor, rigidity, bradykinesia, and postural instability [9,10,11]. Among those symptoms, postural instability can be the most restricting, because it significantly increases the risk of falling and consequently hampers patient independence and quality of life [12,13,14,15,16,17]. In fact, falls due to postural instability are associated with emergency hospitalization and mortality, and considered a strong driver of health care costs [17,18,19,20,21,22,23]. 

Although dopamine replacement therapy can relieve motor symptoms in the beginning, its benefits for postural stability are disputable and its efficacy wears off as the disease progresses [6,15,17,24,25,26,27,28]. Adaptation with gradually higher doses of medication can even induce additional balance-related disabilities, such as dyskinesia or freezing [29,30,31,32]. Therefore, it seems that postural instability is at least partly refractory to dopamine replacement medication, which suggests the involvement of non-dopaminergic circuits in postural control [33,34,35,36,37]. In addition to medication, surgical bilateral high-frequency deep brain stimulation (DBS) of the subthalamic nucleus (STN) can be beneficial for treating PD patients’ abnormal motor symptoms, especially when optimal oral medication fails [38,39,40]. Since STN-DBS affects dopaminergic as well as non-dopaminergic circuits, it has the potential to enhance medication-sensitive symptoms and medically intractable motor disabilities [33,34,35,36,37,41]. Although the mechanisms of DBS are not fully understood, it reduces the excitability of neurons within the STN and consequently normalizes the network interaction between the basal ganglia, thalamus, and cortex [4,29,42,43,44,45,46,47,48]. Studies have reported effective clinical outcomes mainly within the first three years, whereas progressive loss in the long term has been related to disease progression or suboptimal DBS settings, rather than to a lack of DBS efficacy [30,49,50,51,52,53]. The gold standard combines both therapies, effectively and significantly reducing the intake of medication [41,47]. This approach also has synergistic effects on several subdomains of balance. However, studies report divergent results [13,41,54,55,56,57]. 

Some of the divergent results might be due to the methodological limitations of frequently used clinical balance scales and tests. One example is the pull test from the Unified Parkinson’s Disease Rating Scale (UPDRS), which simulates challenges in balance in daily life comparable with sudden bus accelerations or decelerations or slipping and tripping [22,25,58,59,60]. Although this test is easy to apply and is reported to have high sensitivity when performed and interpreted correctly, it might not be the appropriate tool to detect balance deficits early or to analyze the complexity of mechanisms involved in postural control [22,61,62]. Critical aspects of this test include its high inter-rater variability due to the variable pull forces applied to patients’ shoulders, its coarse scoring, and that it only examines postural instability in the backwards direction [60,61]. Moreover, it assesses only the compensatory postural response (CPR), which is the recovery process that occurs after patients have already lost balance. However, anticipatory postural adjustments (APA), meaning preparation in anticipation of an impending balance disturbance, should also be analyzed [17,63,64]. 

A more objective, reliable, and specific analysis can be achieved with standardized balance perturbations in conjunction with electromyography (EMG) and force plates to quantify muscle activation and body sway. Various studies have used computerized perturbations, such as platform translations [12,14,60,63,65,66], rotations [67,68,69], or standardized body pulls [34], to identify Parkinson’s subtypes or for diagnostic purposes including characterizing postural instability compared to healthy subjects. To summarize those study results, PD patients showed inflexibility in controlling muscle activity, including abnormal APA, impaired postural reflexes, and increased agonistic muscle activation with excessive antagonistic co-contractions [65,66,67]. Moreover, patients had difficulties in quickly adapting postural muscle synergies to different perturbation directions, which led to greater sway displacements and longer recovery times [12,14,37,65,66,67,70,71]. However, therapy-induced effects of anti-parkinsonian medication alone and in conjunction with STN-DBS on postural control, such as APA and CPR, have rarely been investigated or compared using such computerized methods. Most studies report limited and insufficient effects of medicinal therapy on balance control [34,63,72,73]. Only minor non-significant improvements in body sway and abnormal muscle activity following perturbations have been reported [34,37,74]. Studies investigating postural instability, comparing it between when DBS was on vs. off and pre vs. post DBS surgery, presented inconsistent results. Turning DBS on revealed immediate improvements in postural flexibility, including enhanced coupling between segmental movements of the body and normalized agonistic contractions during computerized balance perturbations [75,76,77,78]. In addition, comparing postural performance before vs. several months after DBS surgery showed better postural performance and reduced the number of falls [35,78]. In contrast, another study found worse postural performance after DBS surgery with increased sway in the postural preparation phase, which caused a delayed execution of compensatory steps [74]. They even reported that DBS aggravated bradykinesia, which increased the number of falls. Analyzing the effect of combining both therapies on postural instability, the authors found that medication in addition to STN-DBS may have a positive synergistic effect which improves motor control in challenging balance test situations and reduces the risk of falls [35]. However, there are also studies which report that body sway following perturbations did not improve under medication in combination with STN-DBS [79], or was even worse compared to under medication alone [78]. 

Considering all the mentioned study results, the effect of medication and STN-DBS on postural instability in PD seems heterogeneous and unclear. Most previous studies investigating the effect of DBS on postural instability do not include APA or present limited CPR analyses of only forward or backward perturbations. Hence, examining both balance control mechanisms, which are the APA and CPR to multidirectional perturbations, is needed to determine whether or not or why there is a direction-specific risk of falling in PD patients and whether DBS has any benefits. Postural instability is an important criterion for diagnosing and categorizing PD. With this in mind, analyzing APA and CPR to multidirectional perturbations might also be beneficial for counseling patients on appropriate treatment, and could assist clinicians in tailoring and optimizing therapy strategies. This could help to curb the growing economic and emotional burden experienced by PD patients.

Hence, in this study, we aimed to investigate whether PD patients treated with anti-parkinsonian medication in conjunction with STN-DBS have superior postural control compared to patients treated with medication alone. We implemented multidirectional horizontal perturbations of the feet and analyzed force plate and electromyography data to characterize patient APA and CPR. We also tested healthy elderly subjects as a reference. Based on previous study findings, we hypothesized that medication in conjunction with DBS would be advantageous in normalizing patients’ abnormal postural control of APA and CPR. 

## 2. Methods

### 2.1. Subjects

A total of ninety-nine subjects were included in three different study groups: thirty-eight patients with Parkinson’s disease (PD-MED), thirty-one patients with Parkinson’s disease who had undergone previous deep brain stimulation surgery (PD-MED–DBS), and thirty healthy subjects (HS). All PD patients were recruited and tested during a patient consultation at Christian Oehlwein’s Neurological Outpatient Clinic for Parkinson’s Disease and Deep Brain Stimulation in Gera, Germany. The patients were assigned to group PD-MED if they were at least 50 years old and suffered from neurologically diagnosed idiopathic Parkinson’s disease according to the Movement Disorders Society diagnostic criteria with disease severity between 2 and 3 on the Hoehn and Yahr scale [80]. Subjects assigned to group PD-MED–DBS had to meet the same minimum age criterion and to have undergone bilateral deep brain stimulation surgery of the subthalamic nucleus (STN) at least one year prior to ensure optimized DBS settings and full efficiency [72,81,82]. The cut-off for the duration of DBS since surgery was defined as 5 years, since it has been shown that DBS efficacy for axial symptoms, such as postural instability, decreases gradually over time [30,52,53]. All enrolled DBS patients were positive responders to the surgery. The exclusion criteria comprised secondary pathologies affecting the motor and somatosensory systems, causing additional balance dysfunction, for instance atypical parkinsonism, severe camptocormia, normal pressure hydrocephalus, and diabetes mellitus with polyneuropathy. Patients with cognitive deficits (mini-mental state examination (MMSE) <24/30), psychiatric problems, or severe depression were also excluded. Both patient groups were tested in the medication “on” state using regular anti-parkinsonian medication, including levodopa, and patients in the group PD-MED–DBS were additionally in the DBS “on” stimulation state. Clinical data from the most recent neurological examination was provided by the clinic for both patient groups (Table 1). A control group with healthy elderly subjects was examined in the laboratory of the Department of Human Locomotion (Chemnitz University of Technology, Germany). All healthy subjects were free of injuries or diseases, and took no medication that could have interfered with postural performance or cognition. 

### 2.2. Equipment

Postural performance was investigated using a modified version of Posturomed® (Haider Bioswing GmbH, Germany). This device mainly consists of a horizontally mobile platform (60 cm × 60 cm) attached to eight steel cables, which can be locked to reduce platform oscillation [83,84]. For this study, all eight cables were released and the platform was dislocated to 30 mm out of its neutral position and fixed to the frame of the Posturomed device via an electromagnet. A single-axis accelerometer (ADXL78, Analog Devices Inc., Wilmington, MA, USA; sampling rate 1 kHz) was implemented to detect the trigger and the reversal points of the oscillating platform [83,85]. Postural performance was quantified using a force plate (IMM Holding GmbH, Germany; sampling rate 1 kHz) mounted on top of the Posturomed platform [83,85,86]. Additionally, the force plate was customized with a heating system, which provided a constant surface temperature of 27 °C throughout data acquisition to eliminate the influence of fluctuating plantar skin temperatures on postural performance [86]. As shown in a previous study, this setup has good overall reliability [85]. 

The muscle activity of the tibialis anterior (TA) and the gastrocnemius medialis (GM) of both legs was measured using wireless bipolar surface electrodes (TrignoTM Wireless, Delsys Inc., Natick, MA, USA, sampling rate of 1 kHz). EMG electrodes were positioned according to the recommendations of SENIAM, and skin preparation comprised shaving, abrasion with sandpaper, and cleaning with alcohol pads [87,88]. Figure 1 illustrates the total setup.

### 2.3. Testing Procedure and Data Acquisition

Prior to data acquisition, all the subjects were informed about the purpose of this study and provided written informed consent. All the procedures were conducted according to the recommendations of the Declaration of Helsinki and were approved by the ethics committee of the medical faculty of the University Leipzig (IRB number: 023/14-ff). Dynamic postural performance was investigated in four directions with respect to the displacement of the feet: backward, forward, right, and left. The subjects had to adjust their standing direction with respect to the electromagnet for each perturbation direction. For instance, to change from the forward perturbation direction to the left direction, subjects had to turn 90° to the right (Figure 1). Four trials were performed for each direction in a randomized order to minimize the influence of fatigue and habituation [63,66,68]. The subjects were instructed to stand on the force plate in a bipedal stance while barefoot, with an upright posture, keeping the knees straightened but not locked, both arms hanging down loosely, and directing their gaze ahead. To eliminate the effect of stance width on postural performance, both feet of each subject had to be aligned on predefined markers 15 cm apart [12,65]. The platform was released from the electromagnet (trigger) without previous warning and within varying time intervals, initiating unexpected horizontal perturbations with a peak acceleration of 5.7 m/s^2^ for each trial and subject. This perturbation intensity has been tested previously and was chosen to induce challenging in-place perturbations without causing subjects to step or fall [14,69,74,83,85,86,89]. The total duration of data acquisition for each trial lasted 7 s, including a 2 s pre-trigger and 5 s post-trigger interval. For each trial, subjects were instructed to maintain balance while data were recorded simultaneously using a routine written in LabView 8.0. All the subjects were secured with a safety harness by an assistant standing nearby. 

### 2.4. Data Processing

All data were processed using a routine written in MATLAB R2020a (Math-WorksTM, Natick, MA, USA). The accelerometer data underwent offset correction and were filtered using a recursive zero-phase-shift filter with a cut-off frequency of 35 Hz and used to calculate the perturbation trigger. The perturbation trigger was defined as the moment when the accelerometer signal was greater than three times the standard deviation from a 250 ms time interval before the trigger. The accelerometer data were also used to calculate the first two reversal points of the oscillating platform to define the time intervals at which EMG data would be analyzed. 

For the EMG data analysis, three fixed time intervals were used to analyze APA and CPR in relation to the trigger: pre-trigger interval: −250 ms to 0 ms (trigger); post-trigger interval 1: 0 ms (trigger) to +250 ms (reversal point 1); and post-trigger interval 2: +250 ms (reversal point 1) to +485 ms (reversal point 2) (Figure 2). The EMG data underwent offset correction, were rectified, and band-pass filtered (20–500 Hz; Butterworth 4th order). Time to muscle activation for the GM and TA was defined as the moment at which the amplitude was at least 25 ms, greater than three times the standard deviation from the 250 ms pre-trigger time interval. Since early lower extremity responses to perturbations can occur with short latencies of approximately 50 ms, searches were performed in an interval between +25 ms to +485 ms (reversal point 2) post-trigger [90,91,92]. Muscle activity was quantified by the root mean square (RMS) and normalized to the maximum amplitude for each trial on the same muscle and subject [93,94]. Additionally, the co-contraction ratio, which is a measure of inter-muscle coordination and joint stiffness, was calculated as the percentage between the GM and TA using the mean of the normalized EMG data [66,78,95,96]. 

Force plate data were filtered with a recursive zero-phase-shift filter using a cut-off frequency of 50 Hz and used to calculate the center of pressure (COP) ranges for two fixed time intervals with respect to the trigger: the pre-trigger interval total was −2 s to 0 s (trigger) and the post-trigger interval total was 0 ms (trigger) to +5 s. The COP ranges were analyzed in two directions with respect to the perturbation direction. The COP range towards the direction of the perturbation was defined as the primary COP range and the COP range perpendicular to the perturbation direction was defined as the secondary COP range (Figure 2). The COP range represents the maximal COP displacement and is a measure of the limits of stability under which subjects can safely sway without losing balance [60,81,97,98].

### 2.5. Statistical Analysis

For the statistical analysis of the EMG and COP data, the mean of the last three trials on each subject and group was used to reduce the influence of fatigue and habituation [63,66,68]. Trials were excluded in which subjects accidently took a step. All EMG signals were visually inspected, and trials were discarded in which the onset of the muscle activation could not be clearly identified. Normal distribution was checked using Shapiro–Wilk tests (α = 0.05). Differences between the study groups were analyzed using a one-way analysis of variance for the normally distributed data and Kruskal–Wallis tests for the non-normally distributed data. 

The level of significance of the demographic and clinical data was Bonferroni-corrected, depending on the number of study group comparisons (α = 0.05/3 = 0.0167). The level of significance of the EMG and COP data was Bonferroni-corrected (α = 0.05/12 = 0.0042) to account for the study groups and testing conditions. Additionally, the effect sizes, r, were calculated. To analyze intra-group variability, an overall coefficient of variation was calculated for each study group for each parameter in each direction of perturbation. 

Intra-group comparisons between the left and right sides of the body for the normally distributed EMG data were performed using the *t*-test for paired samples and the Wilcoxon–test for the non-normally distributed data (α = 0.05).

## 3. Results

### 3.1. Demographic and Clinical Data

As shown in Table 1, there were considerably more male than female subjects in all the study groups. On average, patient group PD-MED–DBS comprised younger subjects compared to both of the other groups, with statistically significant differences compared to the HS group. Self-rated balance confidence was lower for both patient groups compared to the healthy subject group, HS. No differences were found for the clinical data MMSE, UPDRS III, UPDRS total, or Hoehn and Yahr ratings between the groups PD-MED and PD-MED–DBS. On average, the PD-MED–DBS group suffered from Parkinson’s disease for more than twice as long as the PD-MED group. The disease-dominant side of the body was equally distributed for the PD-MED group. However, the PD-MED–DBS group comprised more patients with disease dominance on the right side of the body. In the PD-MED group, the time interval between the most recent clinical examination and the perturbation tests was longer and the variability was higher compared to the PD-MED–DBS group. The average duration of DBS since surgery was 27.8 ± 10.3 months.

### 3.2. EMG

The intra-group comparisons did not reveal any differences between sides for any EMG parameter and for the backward or forward perturbation directions. Consequently, data from the left and right sides of the body were pooled together and averaged for the backward and forward perturbation directions for each subject in each study group, respectively [68,99]. 

#### 3.2.1. Muscle Activity

Depending on the muscle, time interval, and perturbation direction, several statistically significant differences were found between study groups (Figure 3). Generally, both groups, PD-MED and PD-MED–DBS, showed significantly higher muscle activation at the pre-trigger interval and both post-trigger intervals compared to the HS group. Only a few significant differences between the groups PD-MED and PD-MED–DBS were found for TA at post-trigger interval 2. The overall coefficient of variation for muscle activity showed higher group variability for the groups PD-MED (0.66 ± 0.35) and PD-MED–DBS (0.48 ± 0.19) compared to the HS group (0.39 ± 0.17). The intra-group analysis revealed different levels of muscle activation between the left and the right sides of the body for each group and time interval, mainly of the GM muscle. For example, for the pre-trigger interval, the anticipated perturbations of the subjects’ feet to the right led to higher muscle activity during APA on the right side of the body, just as perturbations to the left led to higher muscle activity during APA on the left side of the body. For both post-trigger intervals, perturbations of the subjects’ feet to the right led to CPR with higher muscle activity on the left side of the body, just as perturbations to the left led to CPR with higher muscle activity on the right side of the body. 

#### 3.2.2. Co-Contraction Ratio

The main result of the inter-group analysis was that both study groups PD-MED and PD-MED–DBS showed muscle co-contractions with higher antagonist and lower agonist ratios compared to the HS group (Figure 4). Nevertheless, the group PD-MED–DBS tended to have less abnormal co-contraction ratios compared to the group PD-MED. The overall coefficient of variation revealed lower group variability for the groups PD-MED (0.26 ± 0.10) and HS (0.27 ± 0.13) compared to the group PD-MED–DBS (0.30 ± 0.11). 

#### 3.2.3. Time to Muscle Activation

Numerous statistically significant differences between study groups were found only for the GM (Figure 5). The main findings were that there was earlier GM muscle activation in the group PD-MED–DBS, primarily under the backward and forward perturbations, and delayed GM muscle activation in the HS group under the forward, right, and left perturbations. The overall coefficient of variation for the parameter of time to muscle activation revealed higher variability for the groups PD-MED (0.25 ± 0.14) and PD-MED–DBS (0.24 ± 0.15) compared to the HS group (0.19 ± 0.10). The intra-group analysis revealed different muscle activation latencies between the left and the right sides of the body under the left and right perturbations for each study group, but only for the GM. For example, when the subjects’ feet were perturbed to the right, the left GM muscle contracted earlier than the right GM muscle. The same pattern but reversed was found for perturbations to the left. 

### 3.3. COP Range

Comparing the study groups, statistically significant differences were found at the total pre-trigger and total post-trigger intervals, however only for the secondary COP ranges (Figure 6). The overall coefficient of variation of the COP range revealed higher group variability between the groups PD-MED (0.41 ± 0.14) and PD-MED–DBS (0.40 ± 0.14) compared to the HS group (0.34 ± 0.15). Between all four perturbation directions, no considerable differences were found for the primary COP range in any study group. 

## 4. Discussion

Maintaining postural stability during external perturbations mainly requires two adequately functioning mechanisms provided by the central nervous system: anticipatory postural adjustments and compensatory postural responses [34,63,64,65,66,74,85,86,95,100,101]. 

### 4.1. Anticipatory Postural Adjustments

Anticipatory postural adjustments are generally associated with the activation and inhibition of the trunk and leg muscles to minimize the negative consequences of forthcoming destabilizing forces [17,63,64]. In particular, when the perturbation direction is predictable, anticipatory strategies, such as leaning and shifting the body weight through the pre-activation of selected leg muscles, indicate preparation and intact voluntary postural control [63,64,67,73]. Depending on the perturbation direction, those postural strategies were found in all three of our study groups. For example, all groups showed pronounced pre-activation of the GM muscles prior to the forward perturbations, which could presumably have been caused by subjects leaning the trunk towards the anticipated forward perturbation to compensate for the counteracting COP shift backwards after the perturbation [67]. This strategy was even more accentuated for the perturbations to the left and right, due to the higher pre-activation of leg muscles on the side of the body toward which the subjects’ feet were perturbed. 

Even though the groups PD-MED and PD-MED–DBS showed APA, their characteristics differed from those of the healthy subjects. The most striking differences were noticed for pronounced TA muscle activity, with higher intra-group variability and abnormal muscle co-contraction ratios being found, which also have been found in other studies [65,67]. Those results might be explained in part by the interference of other early parkinsonian motor symptoms, like rigidity and bradykinesia, which are associated with hypertonic muscle contraction and increased background muscle activity [63,65,102,103,104]. Although there were no obvious signs of significant posture deformities in our patient cohorts, stooped parkinsonian posture with slightly flexed knees and a flexed trunk may have increased background muscle activity as well [76,91,105,106]. Increased background muscle activity, in turn, increases the co-contraction of agonistic and antagonistic ankle muscles and therefore leads to joint stiffness [65,66,67]. Furthermore, since both of our patient groups reported significantly lower self-confidence in balance, additional co-contraction may have resulted from increased voluntary muscle activity due to fear of falling [67]. The effect of fear causing higher background muscle activity was measured in subjects who stood on an elevated platform or when the dimensions of the base of support were changed [60,65,107,108,109,110].

Since it is generally known that muscle activity correlates with COP displacement, the increased background muscle activity of patients might help in the interpretation of the results we found for the COP ranges [111,112]. Knowing that postural instability in PD can be characterized by extensive postural sway, we expected patients to have increased COP ranges compared to healthy subjects, especially towards the anticipated perturbation directions, which we defined as the primary COP range [14,17,60,66,72,108,110,113]. Instead, we only found significantly higher values for both patient groups compared to the healthy subjects for the COP ranges perpendicular to the anticipated direction of perturbation, defined as the secondary COP range. Although those results were surprising, there are some possible explanations. For example, APA, such as leaning towards the perturbation direction with pronounced quasi-isometric antagonistic muscle co-contractions and joint stiffness, might have helped stabilize the patients’ posture, especially towards the anticipated direction of perturbation [67,96,114]. However, those mechanisms may not normalize patients’ postural instability perpendicularly to the direction of perturbation. Similar results have been found in other studies [74,101,115,116]. Even though these studies investigated compensatory stepping initiated by backward perturbations or voluntary step initiation rather than in-place perturbations, they reported that PD patients had difficulties inhibiting APA due to abnormal impulsive behavior and defective postural preparation. This may have caused prominent lateral weight shifts perpendicular to the direction of perturbation [74,101,115,116]. Moreover, PD patients’ extended secondary COP ranges could have been additionally emphasized by tremor [103], or -an asymmetry of motor symptoms, which is typical in Parkinson’s disease [24,69,109,117].

Comparing the PD-MED and PD-MED–DBS groups with each other, we found that APA tended to be more abnormal in the patient group treated with anti-parkinsonian medication alone. This can be seen in the muscle activity and co-contraction ratios. The explanations for this are related to the mechanisms of the levodopa medication and its limited effects on specific structures within the brain. While the lack of dopamine has been considered the main pathophysiological feature of the disease, there is evidence that postural instability in PD is also caused by the impairment of non-dopaminergic circuits, such as the cholinergic system [33,34,35,36]. In this regard, it has been shown that cholinergic cell loss and lesions within the pedunculopontine nucleus cause a decrease in thalamic cholinesterase activity, which is associated with the progressive deterioration of postural instability and an elevated risk of falling as the disease evolves [32,118,119,120,121,122,123]. Consequently, dopaminergic medication may relieve dopamine deficiency symptoms, but has no effect on the postural dysfunction related to the cholinergic system [37,56,123]. Furthermore, postural instability becomes increasingly refractory to medical treatment over time, and adaptation to continually higher doses of medication might aggravate additional balance-related disabilities, like dyskinesia or freezing [29,30,31,32]. Thus, among the cardinal motor symptoms of PD, previous studies have reported that axial symptoms, including postural control, generally respond poorly to levodopa medication [41,63,73,77,124,125]. Other than May et al., who found positive levodopa-induced effects on APA using a rather subjective balance evaluation test, most studies have reported the limited and insufficient effects of levodopa on axial symptoms, such as balance control and APA [63,72,73,77,124]. For example, Schlenstedt et al. found that there were no beneficial effects of levodopa on impaired parkinsonian APA prior to self-initiated perturbations [63]. Furthermore, the study by Hall et al. showed that administering levodopa failed to adapt APA to novel or familiar changes in external postural supports [73]. 

In contrast to levodopa medication, STN-DBS impacts both dopaminergic and non-dopaminergic circuits. In addition, it shows the potential to enhance medication-sensitive symptoms and medically intractable motor disabilities [33,34,35,36,37,41]. Although the exact mechanisms of DBS remain elusive, delivering a high-frequency current to the STN in the basal ganglia causes a complex pattern of inhibitory and excitatory effects, which modulate the entire network between the basal ganglia, thalamus, and cortex. The current explanations include various mechanisms, including synaptic inhibition and depression, the depolarization blockade, the stimulation-induced disruption of pathological network activity, and the stimulation of afferent axons projecting to the STN [4,29,41,42,43,44,45,46,47,54,126,127,128]. Another hypothesis is that the high-frequency signal of DBS overwrites pathological spike train patterns, which causes dysfunction within the basal ganglia–thalamo–cortical and brainstem motor circuit [48]. Other than St. George et al., who reported negative effects on the preparation phase of compensatory steps after turning the STN stimulator on [74], the majority of studies investigating APA report beneficial effects [49,74,100,115,129]. Several studies investigating APA prior to voluntary or compensatory stepping following perturbations found DBS-induced reductions in imbalance with normalized COP displacements, significant improvements in the vertical alignment of the trunk and shank, and reductions in abnormal tonic activity in leg muscles [49,74,77,115,129,130,131]. This suggests that DBS affects the preparation phase, while primarily having an impact on movement amplitudes, which is an indicator of improved postural control in PD patients treated with STN-DBS [49,50,74,100,115,129]. 

Usually, both therapies are used simultaneously, and DBS is accompanied by administering levodopa, the dosages of which, however, can be significantly reduced. Although STN-DBS and levodopa treatment act on different neurological mechanisms underlying postural control, several studies have reported that both therapies in tandem might have a synergistic effect on several subdomains of balance adjustments by modulating the non-dopaminergic descending STN-pedunculopontine nucleus pathway [41,54,72,130,132,133,134]. Bejjani et al. found that STN-DBS and levodopa in combination improved total motor ability by about 80% compared to suprathreshold doses of levodopa alone [54]. Other studies also reported that the bilateral, high-frequency stimulation of the STN ameliorates levodopa-induced motor complications and that axial symptoms, which are unresponsive to levodopa or resistant to preoperative levodopa therapy improved after surgical intervention [41,55,135,136,137,138,139]. Reducing medication dosages in combination with DBS might also be a major factor in reducing levodopa-induced postural side effects [41,47]. Nevertheless, there are also studies that have reported the opposite results [13,56,57,140]. Yin et al. and De la Casa-Fages et al. concluded that STN-DBS only improved balance performance without medication [13,56], and McNeely et al. found no results indicating the synergistic effect of medication and STN-DBS [140]. Rocchi et al. stated that DBS might have attenuated the negative effects on postural sway that were introduced by levodopa [57]. In our study, we also found limited beneficial effects of medication in conjunction with STN-DBS on APA compared to treatment with medication alone.

### 4.2. Compensatory Postural Responses

#### 4.2.1. Timing of Compensatory Postural Responses

Once the mobile platform was released from the electromagnet, the compensatory postural responses of agonistic and antagonistic muscle contractions of appropriate timing and magnitudes were necessary to restore body displacement to maintain stability throughout the perturbations. For example, during the forward perturbations of the subjects’ feet, the agonistic TA muscles had to contract prior to the antagonistic GM muscles within post-trigger interval 1 to produce a forward stabilizing ankle torque that compensated for the counteracting posterior COP shift and that helped to maintain the COP within its limits of stability [60,64,66,78,81]. While several studies report delayed CPR in PD patients compared to healthy subjects following mechanical horizontal perturbations [66,70], sudden ankle rotations [37,92,141,142], or electrical nerve stimulation [70,143,144], the results of our study and two other studies do not confirm delayed muscle activation latencies in PD patients [65,67]. This contradiction may be due to the fact that reflex abnormalities, including delayed latencies, occur predominantly in more severe stages of the disease, while the patients in our study had milder stages of PD [17,66,70]. 

Nevertheless, one of our main findings was the early occurrence of antagonistic muscle activation in both patient groups compared to the healthy subjects group. This was especially pronounced for the perturbation directions forward, right, and left. For example, for the forward perturbations of the subjects’ feet, the GM muscles in both patient groups co-contracted as antagonistic muscles together with the agonistic TA muscles within post-trigger interval 1, instead of becoming active after the first reversal point within post-trigger interval 2. This unphysiological early muscle co-contraction causes stiffness and reduced joint mobility with inflexibility to react and adapt to changing perturbation directions. This can be obstructive, since the perturbation direction in our study was inverted after the first reversal point [12,65,66]. The timing and magnitude of compensatory postural reactions to perturbations depend on the intactness of the integration of various afferent information, including that of the visual, vestibular, and somatosensory systems. Therefore, defective afferent input from these systems in PD may help explain the patients’ abnormal early antagonistic muscle co-contractions [145,146,147]. Especially during fast perturbations of the feet, postural reactions rely heavily on proprioceptive feedback steming from muscle spindles, Golgi tendon organs, and joint receptors of the lower extremities, as well as mechanoreceptors of the plantar feet, providing information about the perturbation intensity and direction [69,86,148,149]. The disease-related false interpretation of this afferent information might have been caused by defective impulsive motor behavior and reflex inhibition, and consequently could have caused the patients’ abnormal early antagonistic co-contractions [116,150,151,152]. This may be because this afferent information is involved in scaling reflex responses through various synaptic activation and inhibition mechanisms, such as antagonistic and autogenic inhibition, which are also thought to be impaired in PD [150,151,152,153]. Another explanation may have to do with the postural strategy that PD patients use mainly to stabilize the body to keep the COP within the limits of stability following perturbations. The body is multi-segmented with various main articulation points, such as the ankle, knee, and hip joints. Therefore, there are basically two different postural strategies, namely the ankle strategy and the hip strategy, which are characterized by the coupling of body segments [65,75,154,155]. Since increased muscle tone and rigidity in PD also increases the coupling between body segments, PD patients are forced to use the ankle strategy instead of the hip strategy, which is a rather inefficient strategy for preventing falls [65,66,102]. Due to the long lever, the body acts as a reverted pendulum, which can cause PD patients to fall like a log [50,65,67]. The ankle strategy involves the earlier activation and pronounced activity of muscles around the ankles, which may have caused the early antagonistic co-contractions in our PD patient groups.

Another main finding of our study is of the occurrence of early muscle activation in the stimulated patient group PD-MED–DBS compared to the other two study groups, PD-MED and HS. To the best of our knowledge, hardly any other study has investigated the effect of STN-DBS on mechanically-induced postural reflex latencies. Only St George et al. published data about muscle activation latencies induced by horizontal forward perturbations of subjects’ feet. By comparing different therapy conditions, they found comparable initial EMG bursts of the TA muscle at approximately 100 ms for each study group, regardless of the medication or DBS state [78]. Due to the lack of agreement with our findings, we wondered what could have caused the STN-MED–DBS patients in our study to have earlier muscle activation. We visually inspected each individual EMG trial and checked the onset of muscle activation that was detected by our algorithm. Therefore, a methodological bias should have been excluded. However, Patel et al. investigated the effect of STN-DBS in the “on” vs. “off” state on patients’ postural strategies following balance perturbations. While they found no change in postural strategy with DBS in the “off” state, there was an increase in coupling between segments for DBS in the “on” state. Switching DBS on increased coupling between segments, including the ankles, which may have simplified the corrective postural responses and consequently have caused the earlier and more pronounced activation of lower leg muscles [75]. This is supported by several other studies that found that STN-DBS alters postural alignment and consequently can effect postural reflex responses [71,76,105]. Studies investigating the latency of electrically induced so-called Hofmann (H) reflex, which is an analogue of the mechanically induced stretch reflex, also provide evidence that STN-DBS may change reflex timing [156,157,158]. Pötter-Nerger et al. reported the facilitation effects of the H-reflex after STN-DBS was switched on [156]. Using transcranial magnetic stimulation, Hidding et al. found that motor-evoked potential onset latencies were significantly shortened by STN stimulation [158]. Another reason for the occurrence of early muscle activation in our DBS group might be the impact STN-DBS has on inhibition mechanisms, such as antagonistic inhibition and autogenic inhibition, which are involved in modulating reflex timing and amplitude [70,153]. The muscle latencies found in our study mostly occurred after 50 ms. Consequently, postural reflexes should have comprised mainly medium and long latency responses, which are predominantly under supraspinal control including the network between the basal ganglia, thalamus, and cortex [90,92,142]. Stimulation of the STN can spread out and affect several structures in close proximity that play an important role in balance function, such as the pedunculopontine nucleus and the mesencephalic locomotor region. Therefore, a co-stimulation of those structures may have led to the facilitation of reflex latencies through descending corticospinal or reticulospinal pathways [4,29,30,36,41,42,43,44,45,46,47,54,69,118,126,127,128,158,159,160].

#### 4.2.2. Magnitude of Compensatory Postural Responses

When analyzing the magnitude of CPR, we found higher muscle activity with pronounced antagonistic co-contractions after the perturbation trigger for both patient groups compared to healthy subjects HS. Hence, our results are in line with those of most other studies that reported exaggerated amplitudes, especially for medium or long latency responses. [37,65,67,78,91,92,142,161,162]. We found equivalent EMG characteristics with higher muscle activity for patient APA prior to the perturbation trigger. Therefore, the influence of other parkinsonian motor symptoms, impaired sensory integration, defective impulsive motor behavior and inhibition mechanisms, and increased fear of falling may have also provoked pronounced antagonistic muscle co-contraction with increased joint stiffness at both post trigger intervals [63,65,66,67,102,103,104,150,151,152]. The perturbations in our study caused brief arm-raising reactions to counteract body displacement. Therefore, the delayed and ineffective arm movements of the PD patients may have been another reason the leg muscles compensated with higher activity [67,68,100,110]. Increased muscle activity may have also been a compensation mechanism for patients’ diminished muscle force, which has been related to the presence of less ankle torque and ineffective counter movements [34,67,69]. Moreover, the early occurrence of antagonistic muscle activation we found in both patient groups might have additionally facilitated higher EMG RMS values, especially for post-trigger interval 1. 

It has been shown that patients’ abnormal muscle activity can translate into extensive COP sway and higher COP velocities following perturbations. Therefore, we also assumed increased COP ranges for our patient groups compared to healthy subjects, especially towards the perturbation directions, which we defined as the primary COP range [12,14,17,21,34,60,65,66,72,78,98,108,110,113]. However, we could not confirm our assumption, as we did not detect any group differences between the primary COP ranges. It seems that our patients’ pronounced muscle co-contraction might have served as a balance strategy, because joint stiffness increases movement resistance and consequently diminishes sway amplitudes [14,65,67,163,164]. This is confirmed by two other studies that also found equal or even smaller COP ranges for PD patients compared to healthy subjects after horizontal perturbations [12,14]. Nevertheless, we found significant differences between our healthy subjects and both patient groups for COP ranges perpendicular to the perturbation direction, defined as the secondary COP range. Since we found similar results for the pre-trigger interval, we presume that the pre-trigger COP sway perpendicular to the direction of perturbation was amplified after the platform was released from the electromagnet, because it was free to oscillate within both degrees of freedom. The extended secondary COP ranges of the PD patients could have been additionally caused by an asymmetry of motor symptoms, which is typical of Parkinson’s disease, and by the presence of less muscle force, which relates to diminished sway resistance [24,34,69,109,117].

Previous studies have shown that postural instability in PD can be dependent on the perturbation direction. Therefore, we aimed to analyze balance performance using multidirectional perturbations to determine whether or not and why there may be a direction-specific preponderance of postural instability [12,60,65,67]. Based on biomechanical principles, it is generally known that upright standing humans are most unstable under backward body displacements, because the short lever of the calcaneus is less effective at generating sufficient ankle torque compared to the longer lever of the forefoot [165,166]. Another reason why backward sway is biomechanically more unstable is because the COP in these situations is already situated very close to its limits of stability [167,168]. Direction-specific postural instability in PD might especially be amplified by the neural constraints caused by the disease [12,60,65,67]. Other studies have reported that patients predominantly suffer from impaired postural instability with higher COP displacements under backward and lateral body displacements, caused by antagonistic muscle co-contractions, which reduce joint mobility of the knees, hips, and trunk [12,60,65,67]. In our study, however, we found no direction-specific balance abnormalities between patients and healthy subjects. 

We found mainly subtle differences when comparing the CPR between both of our patient groups, however patients of the PD-MED group showed a tendency to have worse postural control. Levodopa medication restores dopaminergic pathway functionality within the basal ganglia by decreasing the negative output of the thalamus to the motor cortex. This explains improvements in akinesia and rigidity, but its effect on postural instability remains under debate. [4,5,6,26]. As already mentioned, this might be because that postural impairments in PD patients are less associated with dopaminergic lesions and more associated with pathological processes beyond the dopaminergic system, such as the cholinergic systems, including the pedunculopontine nucleus [6,32,33,35,36,37,118,120,121]. This may explain why several studies have reported that levodopa medication generally shows little effect on correcting abnormal postural reflex responses, muscle amplitudes, or postural sway following perturbations [13,34,37,74,78,81,125]. Bloem et al. only reported partial improvements in COP displacements, and there was a failure to correct medium latency reflex amplitudes [33,34,36,37]. Furthermore, Di Giulio et al. and St. George et al. reported that levodopa had no significant effects on either in-place postural responses or protective stepping deficits following perturbations [34,74,78]. Another study comparing the effects of medication revealed increased sway velocity in the medio-lateral direction, as well as increased sway amplitude on medication [169]. 

Studies comparing levodopa medication to STN-DBS have reported that even the best medication setting produces less postural improvement than DBS does [39,40]. Those DBS benefits might be associated with the influence of dopaminergic and non-dopaminergic pathways of postural control through projections from the STN and thalamus to the pedunculopontine nucleus [33,34,35,36,37,41,118]. It is thought that the stimulation of the STN modulates the network between the basal ganglia, thalamus, and cortex, and restores the functionality of those systems, while the pedunculopontine nucleus might play the role of a relay station providing the basal ganglia with information for posture modulation [4,29,41,42,43,44,45,46,47,54,126,127,128]. Nevertheless, study results regarding the effect of STN-DBS on CPR are controversial. Studies that found beneficial effects of the former comparing DBS in the “off state” with DBS in the “on” state reported enhanced postural strategies, improved body position, and increased agonist muscle burst durations during balance perturbations [75,76,78]. Using rather subjective clinical balance tests, Li et al. reported positive effects on postural instability after DBS surgery, which were still noticeable after 12 months [81]. On the other hand, some studies reported insufficient effects of STN-DBS on CPR. For example, St George et al. stated that STN-DBS did not improve the characteristics of compensatory steps following perturbations. In fact, STN-DBS delayed step execution and altered leg muscle response amplitudes. This, however was argued to be a consequence of the disrupted postural preparation phase. They concluded that the detrimental effect of DBS was greater than the benefit of it [74]. In the studies by Patel et al. and May et al., the authors also concluded that DBS was not able to restore adaptive motor control abilities in PD patients [72,170]. 

As mentioned above, DBS together with medication is the common method of treatment, and can have positive effects [35,54,72], no effect, or even negative effects on postural control [13,78,79,171]. For example, May et al. found that levodopa together with DBS may affect balance in various dynamic situations due to the presence of better-integrated sensory feedback [72]. Colnat-Coulbois et al. also showed that STN-DBS in combination with levodopa treatment reduced postural instability by increasing motor abilities and specific posture-related mechanisms that lead to a reduction in falls. They reported improved balance precision in challenging balance test situations and improved postural strategies. Patients were able to adapt their balance more accurately in situations with sensorial conflicts. They concluded that STN stimulation combined with levodopa treatment allowed the basal ganglia to function again, because the stimulation of the STN influences the dopaminergic and non-dopaminergic pathways separately [35]. However, since our study found only subtle differences between both patient groups, our results seem to be in line with studies that reported less beneficial effects under treatment with STN-DBS in conjunction with medication [13,78,79,171]. For example, St George et al. showed that CPR following backward body displacements were worse under the condition of treatment with medication combined with STN-DBS compared to that of medication alone [78]. Similarly, Yin et al. and Benabid et al. reported that balance improvements were only measurable without medication [13,171]. Additionally, in the study by Maurer et al., abnormal COP parameters following body tilt perturbations did not improve under the condition of treatment with medication in combination with STN-DBS [79]. 

As our study findings for APA and CPR show rather subtle and limited beneficial effects of treatment with medication in conjunction with STN-DBS compared to medication alone, we questioned what could have caused the lack of STN-DBS efficacy. Therefore, our results should be interpreted in light of the mainly methodological limitations, such as differences in surgical procedures, electrode localization in the STN, and different DBS settings. In fact, as stated by Pötter-Nerger and Volkmann, the lack of STN-DBS efficacy should be distinguished carefully, mainly by a “primary” failure ascribed to suboptimal DBS settings, and by “secondary” failure ascribed to the fading of stimulation-induced benefits due to disease progression [49]. Some of our STN-DBS patients were tested before and some were tested after the neurological consultation. Therefore, the patients who were tested before consultation did not benefit from possibly optimized DBS settings, which could have biased our results towards lower efficacy [172]. The possible fading of stimulation-induced benefits could also have been an issue, since, on average, our STN-DBS patients were affected by the disease 2.4 times longer, and they also had relatively long STN stimulation intervals of approx. 3 years since surgery [30,49,52,53,169,172]. As the subjects in our healthy subject group was on average about 2 to 4 years older than the patients in our patient groups, the effect of aging could also have influenced their postural performance negatively and consequently created bias in our observations [170,173,174]. Further possible explanations include study group composition and individual disease severity. Our patient groups comprised various PD subtypes, such as tremor-dominant, akinetic-rigid, as well as postural instability and gait disturbance subtypes with relatively mild disease severity [14,175]. Due to the patients’ having large amounts of inter-individual characteristics of motor symptoms, a larger sample size might have helped us to detect greater group differences. In addition, a longitudinal interventional study design, comparing postural instability pre vs. post DBS surgery with and without additional medication instead of a cross-sectional design, might have enabled a more precise investigation of which therapy conditions affect certain aspects of postural control. Other methodological limitations were the predictable perturbation directions and that a more intense balance thread with changing and higher perturbation intensities may have been more effective at provoking and revealing direction-specific postural instabilities [69].

## 5. Conclusions

In this study, we investigated whether PD patients treated with anti-parkinsonian medication in conjunction with STN-DBS have superior postural control compared to patients treated with medication alone. We implemented multidirectional horizontal perturbations of the feet and analyzed force plate and electromyography data to characterize patients’ control mechanisms—APA and CPR. For reference, we also tested healthy elderly subjects. Based on previous study findings, we hypothesized that medication in conjunction with DBS would be advantageous in normalizing patients’ abnormal postural control of APA and CPR. In conclusion, we found that regardless of the treatment conditions, patients in both PD patient groups suffered from inadequate APA and CPR with early and pronounced antagonistic muscle co-contractions compared to healthy elderly subjects. Nevertheless, we found no direction-specific preponderances of postural instability in any of our study groups. Comparing the treatment conditions, we only found minor benefits for PD patients treated with medication in conjunction with STN-DBS over PD patients treated with medication alone. The study group PD-MED–DBS showed faster postural reflexes and tended to have more physiological co-contraction ratios, which suggests that STN stimulation might influence the timing and amplitude of muscular control. Nevertheless, because of the lack of significant improvements in the postural stability of patients on anti-parkinsonian medication in conjunction with STN-DBS compared to those on medication alone, we reject our hypothesis. The discrepancy between our study findings with those of studies that showed beneficial effects might have mainly been due to methodological reasons, as described above. Further studies investigating the effect of DBS and medication on postural performance should include varying stimulation parameters of the STN, as well as different combinations of anti-parkinsonian medication. Besides STN-DBS, the effects of the stimulation of other mesencephalic areas, such as the pedunculopontine nucleus and globus pallidus interna, on postural instability should also be analyzed. Perturbations should be unexpected and multidirectional with varying randomized intensities. Further studies should also include EMG analysis of several muscles that stabilize the knee and the hip joints. Moreover, 3D motion analysis would be beneficial for investigating full body kinematics during balance performance.

## Figures and Tables

**Figure 1 brainsci-13-00454-f001:**
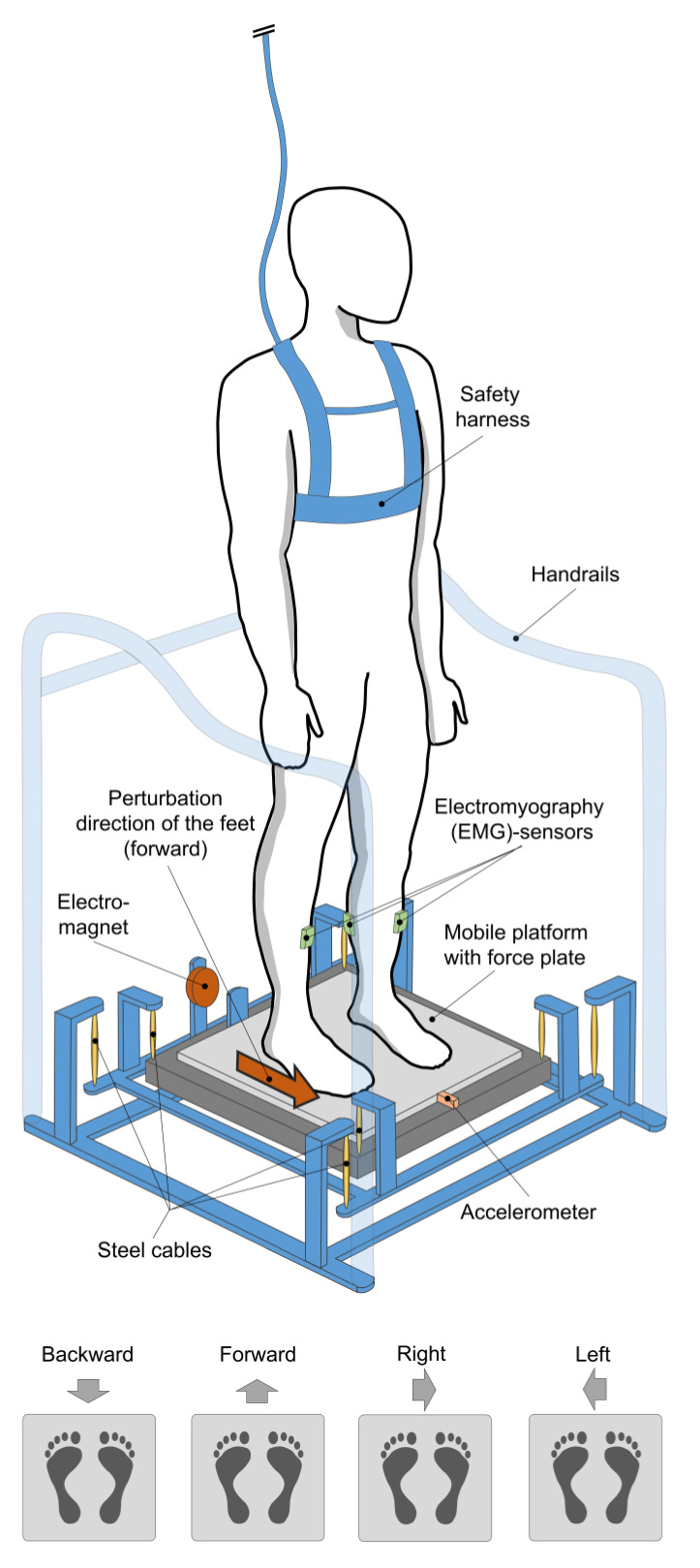
Total setup and all four perturbation directions.

**Figure 2 brainsci-13-00454-f002:**
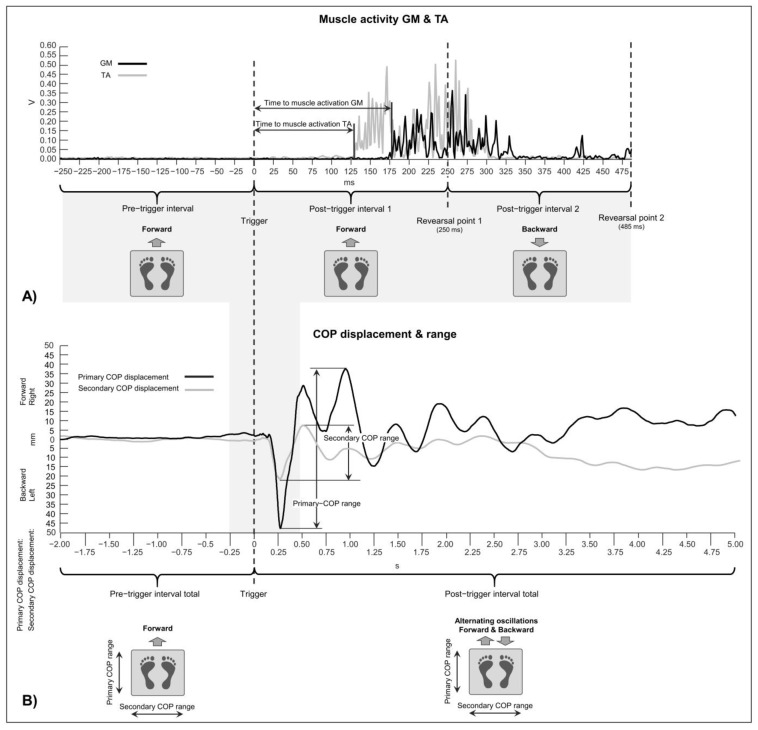
Illustration of an actual electromyography (EMG) muscle activity signal (**A**) and center of pressure (COP) displacement (**B**) from a random patient in study group PD-MED–DBS, who performed a forward perturbation trial. The EMG-RMS data analysis was performed for three defined time intervals (pre-trigger interval and post-trigger intervals 1 and 2). The EMG graphic also visualizes the time to muscle activation with respect to the trigger for the muscles gastrocnemius medialis (GM) (black line) and tibialis anterior (TA) (grey line). Note that the platform movement began with the trigger, and that the initial perturbation direction for post-trigger interval 1 inverted beyond reversal point 1, because the platform swung back. For this example, in post-trigger interval 1, TA worked as an agonistic muscle and GM worked as an antagonistic muscle. This ratio is inverted for post-trigger interval 2. The COP range analysis was performed for the entire 2 s pre-trigger interval total and the 5 s post-trigger interval total. The COP graphic shows the COP displacement and range towards the perturbation direction, which was defined as the primary COP range (black line), and the COP displacement and range perpendicular to the perturbation direction, which was defined as the secondary COP range (grey line).

**Figure 3 brainsci-13-00454-f003:**
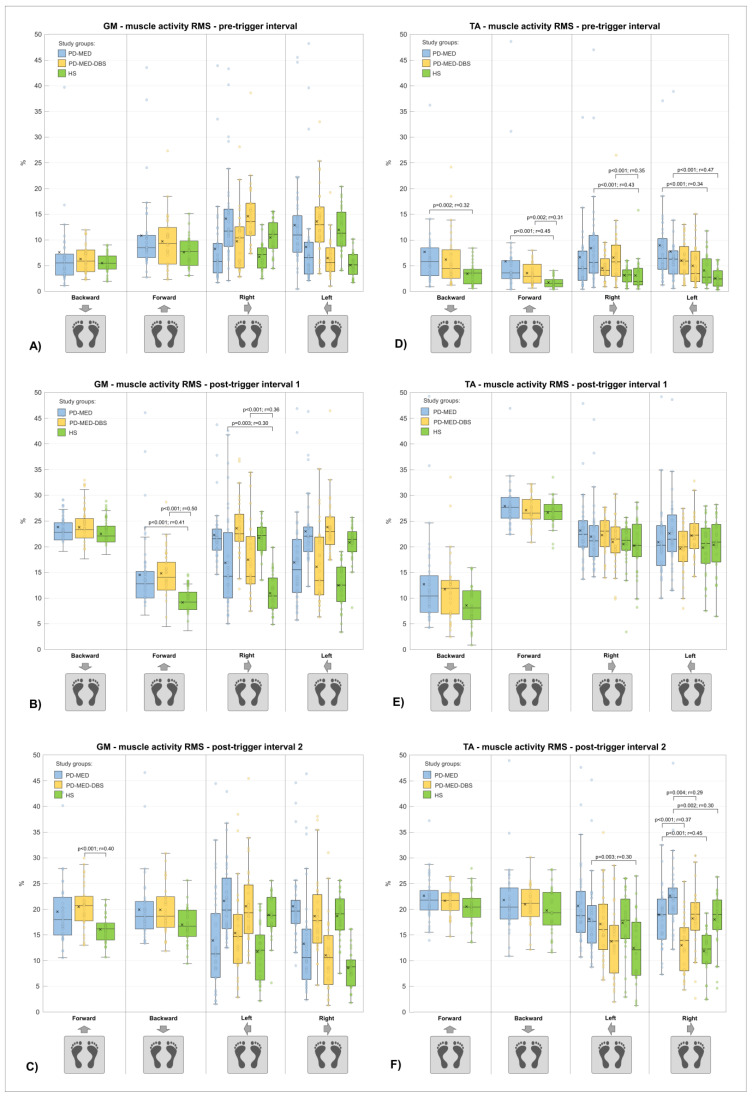
RMS muscle activity of gastrocnemius medialis (GM) (**A**–**C**) and tibialis anterior (TA) (**D**–**F**) of each study group, time interval, and perturbation direction, respectively. Pooled and averaged data of both sides of the body are presented for backward and forward perturbations. Data from the left and right sides of the body are presented individually for right and left perturbations. Statistically significant differences between study groups (*p* < 0.0042) are indicated as well as effect sizes, r. The cross within each box marks the mean value. Note that the platform only moved in post-trigger intervals 1 and 2, and that the initial perturbation directions from post-trigger interval 1 were inverted in post-trigger interval 2 because the platform swung back after reversal point 1.

**Figure 4 brainsci-13-00454-f004:**
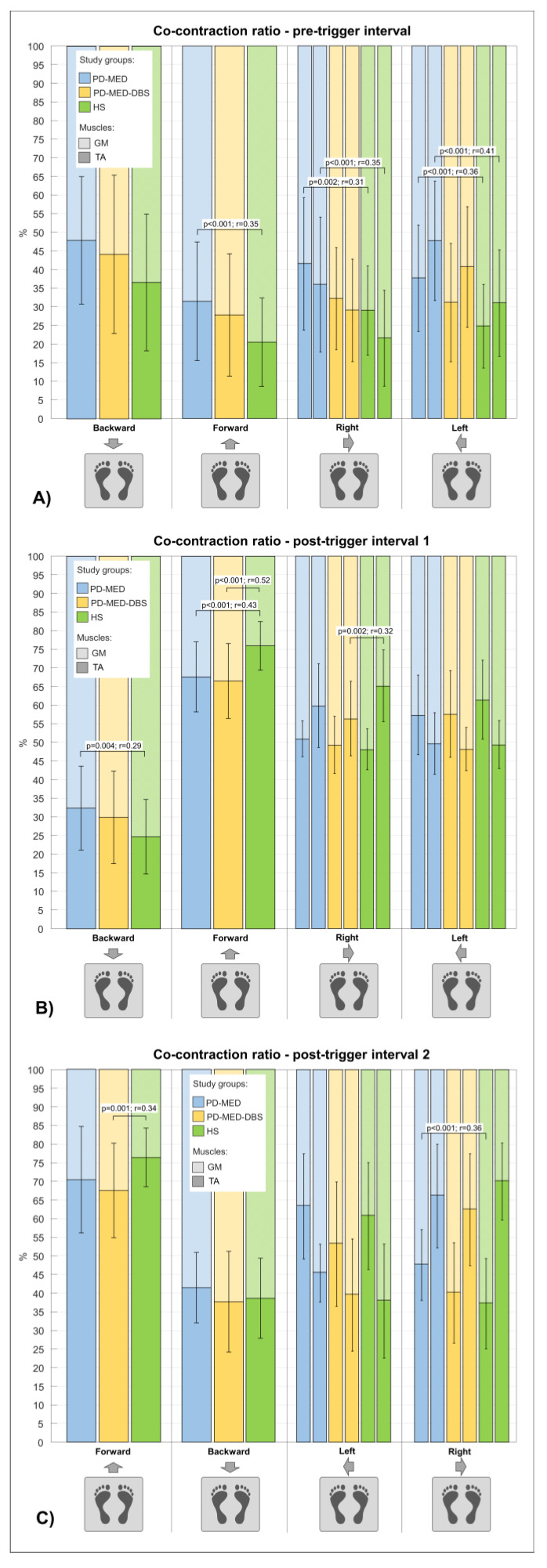
Co-contraction ratio (mean ± SD) between GM (lighter color) and TA (darker color) muscles for each study group, time interval (**A**–**C**), and perturbation direction, respectively. For backward perturbations, the GM worked as an agonist muscle and the TA worked as an antagonist muscle and vice versa under forward perturbations. Statistically significant differences between study groups of *p* < 0.0042 are indicated.

**Figure 5 brainsci-13-00454-f005:**
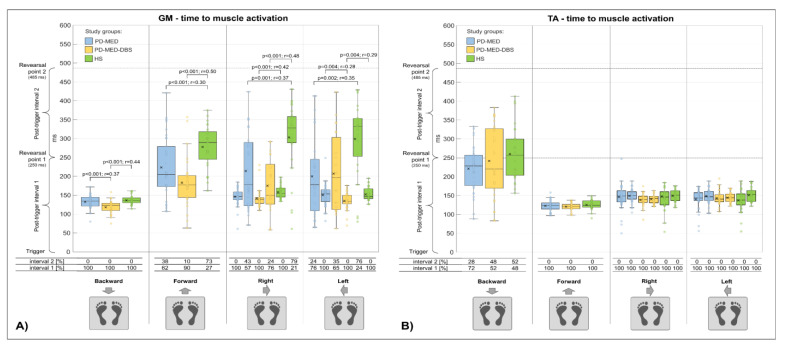
Time to muscle activation of the GM (**A**) and TA (**B**) muscles for each study group and perturbation direction, respectively. The table below the graph shows the percentage of subjects’ muscle activation within post-trigger interval 1 or 2. Statistically significant differences between study groups of *p* < 0.0042 are indicated.

**Figure 6 brainsci-13-00454-f006:**
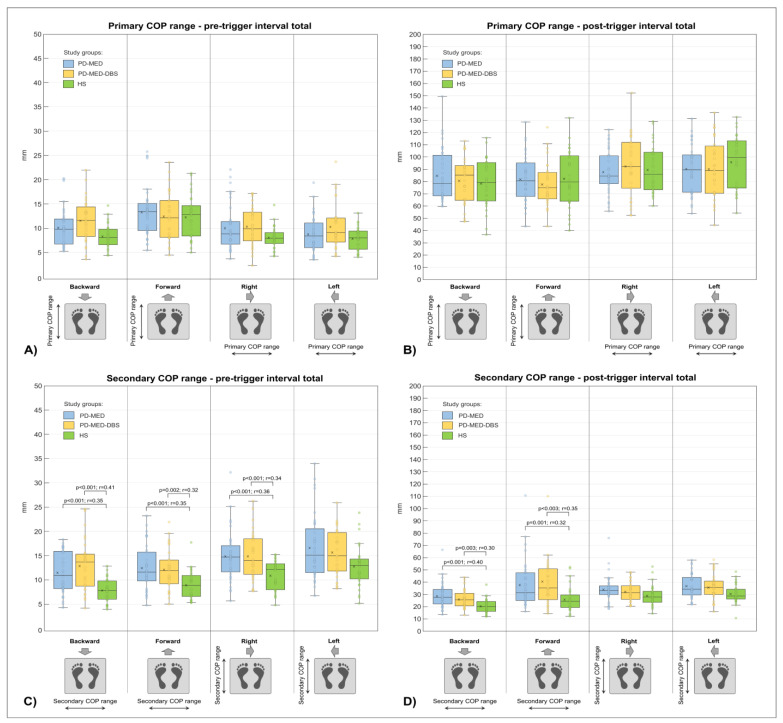
COP ranges of total pre-trigger and post-trigger time intervals for each study group and perturbation direction, respectively. COP ranges towards the perturbation direction are defined as primary COP ranges (top row **A**,**B**) and COP ranges perpendicular to the perturbation direction are defined as secondary COP ranges (bottom row **C**,**D**). Statistically significant differences between study groups of *p* < 0.0042 are indicated.

**Table 1 brainsci-13-00454-t001:** Demographic and clinical data (mean ± SD); statistically significant differences marked with * and #: PD-MED vs. PD-MED–DBS vs. HS, *p* < 0.0167; PD-MED vs. PD-MED–DBS, *p* < 0.05.

		PD-MED	PD-MED-DBS	HS	*p*-Values
demographic data:	n/gender	38/ 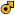 29/ 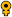 9	31/ 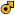 22/ 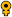 9	30/ 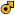 19/ 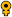 11	
age (years)	68.2 ± 7.6	64.5 ± 7.5 #	70.6 ± 5.7 #	# =0.007
clinical data:	self-rated balance confidence (0–100) (%)	65.0 ± 16.8 *	66.4 ± 17.3 #	80.7 ± 9.4 *#	*# <0.001
MMSE (0–30)	28.6 ± 1.8	28.4 ± 1.8		
UPDRS III (0–108)	15.9 ± 6.7	14.6 ± 5.9		
UPDRS total (0–199)	26.1 ± 10.6	27.5 ± 9.5		
Hoehn and Yahr (0–5)	2.0 ± 0.3	2.0 ± 0.3		
disease duration since diagnosis (months)	78.6 ± 54.6 *	184.4 ± 79.8 *		* <0.001
disease-dominant body side	left: 20; right: 18	left: 11; right: 20		
time between last neurological examination and perturbation test (months)	5.2 ± 14.3	2.7 ± 2.8		
DBS duration since surgery (months)		27.8 ± 10.3		
self-rated satisfaction with DBS (%)		77.9 ± 21.1		

## Data Availability

The dataset used and analyzed in this study is available from the corresponding author upon reasonable request.

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
