# Peer review of "Anticipatory Postural Adjustments and Compensatory Postural Responses to Multidirectional Perturbations—Effects of Medication and Subthalamic Nucleus Deep Brain Stimulation in Parkinson’s Disease"

_brainsci, 2023, doi:10.3390/brainsci13030454_

Round 1
Reviewer 1 Report
The ms by Hess et al. presents well designed and performed research concerning the influence of medical therapy alone or with STN-DBS therapy on the postural reactions in PD. The analyzed groups are comparable, the healthy control is included, the extent of analyzed features is comprehensive.
I would have just 2 remarks/questions:
1. The PD groups are MED and another MED-DBS. Do any analyzable data exist about the postural status of MED-DBS patients before surgery. How surgery changed the status in those patients?
2. The ms need on thorough reading to correct some spelling/grammatical/punctuation/styllistic mistakes.
Author Response
Response 1:
Thank you for your question. Unfortunately, we do not have access to any data on the postural status before surgery. However, as reported by the responsible neurologists all enrolled patients responded positively to the surgery. This means that shortly after surgery they showed significant motor improvements, including postural stability. This has been assessed with the Unified Parkinson's Disease Rating Scale (UPDRS).
Response 2:
Thank you for your comment. The entire manuscript has been proofread by a native speaker. All changes have been marked using the track changes function in MS Word.
Reviewer 2 Report
I read with interest the manuscript Anticipatory postural adjustments and compensatory postural responses to multidirectional perturbations - Effects of medication and subthalamic nucleus deep brain stimulation in Parkinson’s disease. The article is mostly well-written, the introduction is adequate, and the methods are well-described. Likewise, the authors provide an extensive discussion well-supported by numerous references. Finally, their conclusions seem sustained by their results. In this regard, although the results are interesting, the methodology used attracted my attention. I think this approach can provide relevant information to quantitatively evaluate the effects of different treatments on postural instability in patients with Parkinson´s disease (and possibly other diseases that affect movement, balance, etc.).
However, it contains several typos and grammar mistakes. I would suggest a thorough revision before the manuscript is resubmitted.
Author Response
Response:
Thank you for your comment. The entire manuscript has been proofread by a native speaker. All changes have been marked using the track changes function in MS Word.